# Clinicopathologic and prognostic implications of Golgi Phosphoprotein 3 in colorectal cancer: A meta-analysis

**Tao Wang** *, **Jiandong Fei, Shuangfa Nie**

Department of General Surgery, The First Affiliated Hospital of Hebei North University, Zhangjiakou, China

* zjkwangtao@163.com

## Abstract

### Background

Golgi Phosphoprotein 3 (GOLPH3) has been implicated in the development of colorectal cancer (CRC). Nevertheless, the clinicopathological and prognostic roles of GOLPH3 in CRC remain undefined. We thus did a meta-analysis to assess GOLPH3 association with the clinicopathological characteristics of patients and evaluate the prognostic significance of GOLPH3 in CRC.

### Methods

An electronic search for relevant articles was conducted in the PubMed, Cochrane Library, Web of Science, Medline, Embase, CNKI, and WanFang databases. Two independent reviewers searched all the literature and finished the data extraction and quality assessment. Odds ratio (OR) or hazard ratio (HR) with 95% confidence interval (CI) were used to assess estimates. Stata software (version12.0) was employed to analyze the data.

### Results

A total of 8 published studies were eligible (N = 723 participants). Meta-analysis revealed that GOLPH3 was found to be highly expressed in tumor tissues compared to that of adjacent colorectal tissues (OR, 2.63), and overexpression of GOLPH3 had significant relationship with advanced clinical stage (OR, 3.42). GOLPH3 expression was not correlated with gender (OR, 0.89), age (OR, 0.95), positive lymphatic metastasis (OR, 1.27), tumor size (OR, 1.12), poor differentiation of tumor (OR, 0.56) or T stage (OR, 0.70). Moreover, GOLPH3 overexpression was not associated with worse overall survival (OS) (HR = 1.14, 95% CI: 0.42–1.86, $P>0.05$) and disease-free survival (DFS) (HR = 0.80, 95% CI:-0.26–1.86, $P>0.05$).

### Conclusions

GOLPH3 overexpression is correlated with tumor stage, which is an adverse clinicopathological characteristic of CRC. But, GOLPH3 can not serve as a useful biomarker in evaluating the progression of CRC.

**Data Availability Statement:** All relevant data are within the manuscript and its Supporting Information files.

**Funding:** The author(s) received no specific funding for this work.

**Competing interests:** The authors have declared that no competing interests exist.

## Introduction

Colorectal cancer (CRC) is the third leading malignancy worldwide and the second most deadly cancer [1] Patients suffering from CRC in the early stages (I–III) usually have a long survival. Once patients develop malignant metastasis, the 5-year survival rate is less than 13% [2]. Much of the morbidity and mortality of CRC is due to patients being diagnosed in the late stages, where therapeutic intervention is less effective [3]. Additionally, certain biomarkers are important in tumor progression. In this regard, numerous studies have been focused on finding predictive biomarkers for CRC with a special interest in evaluating prognostic significance [4, 5].

GOLPH3, also called GPP34, is located in chromosome 15 p13 [6]. GOLPH3 protein is mainly enriched in the outer membrane on the opposite side of the Golgi vesicle, which also exists in the tubules, vesicles, endohedral chamber and membrane [7]. GOLPH3 is an oncogene that is known to be upregulated in breast cancer, esophageal cancer, glioma, prostate cancer and other cancers [8–12]. GOLPH3 can regulate the function of Golgi protein glycosyltransferase, leading to the abnormal secretion of glycosylated protein, which can affect the growth, adhesion, migration, invasion and immunity of tumor cells [13]. Researches have indexed that GOLPH3 is very relevant to the occurrence, development and prognosis of CRC and may serve as a representative molecular marker [14, 15]. The relationship between the expression of GOLPH3 and the clinicopathological features and prognosis of CRC remains inconsistent and controversial. Therefore, a meta-analysis was conducted to determine the authenticity of GOLPH3 in regard to the clinicopathological features and prognosis of CRC, providing further evidence for clinical practice.

## Materials and methods

### Search strategy

Manual retrieval of PubMed, Cochrane Library, Web of Science, Medline, Embase, CNKI and Wanfang Database (last search updated to December 31, 2020). The following literature were searched from Medical Subject Headings (MeSH) with the following keywords:("Colorectal Neoplasm" or "Colorectal Cancer" or "Colorectal Tumor" or "CRC") and ("Golgi phosphoprotein 3" or "GOLPH3" or "GPP34 protein"). We applied no language restrictions. In addition, references in other related articles were also scanned to find eligible studies.

### Selection criteria

Inclusion criteria are as follows: (1) immunohistochemical detection of GOLPH3 expression; (2) histological diagnosis of the colorectal tumor; (3) Patient clinical information is complete; (4) Language limit: Chinese or English. Exclusion criteria were:(1) Repeat published; (2) The types of studies do not match (editorials, letters, and so on); (3) The patient's clinical data information is incomplete; (4) studies not conducted in humans.

### Data extraction and quality assessment

All articles are screened independently by two independent reviewers to ensure science and fairness. The extracted information included the author, year of publication, country, number of patients, age, pathologic differentiation degree, gender, lymphatic metastasis, tumor stage, tumor size, T stage, hazard ratio (HR) and corresponding 95% CI for OS and DFS. Different studies had different definitions of positive and negative GOLPH3. Accordingly, this paper adopted the definition in the original literature. HRs (95% CI) were extracted directly from

certain studies via univariate and multivariate analyses. If not reported HR and 95% CI, their values were estimated in the Kaplan-Meier curve [16]. The Newcastle-Ottawa Scale (NOS) [17] tool was used to assess the quality of literature. Any disagreement was decided through a discussion of both parties or by consultation with a third team member.

## Statistical analysis

Review Manager5.2 and Stata12.0 software were used for statistical analysis. The correlation between GOLPH3 overexpression and clinicopathological characteristics of CRC was analyzed by OR and 95% CI. Correlation between GOLPH3 overexpression and OS or DFS in CRC was analyzed using HR and 95% CI. According to heterogeneity, this meta-analysis was performed using fixed-effects or random-effects model analyses. HR and 95% CI were extracted using the Engauge Digitizer for references that did not provide HR but provided Kaplan-Meier survival curves instead. The $I^2$ test were used to quantify the potential heterogeneity that may exist among the analyzed studies. In the absence of significant heterogeneity, the data were analyzed by fixed-effect model, otherwise by random effects model.

## Results

### Literature search

The process of article identification, inclusion, and exclusion is shown in Fig 1. Based on the defined criteria, 284 potential studies were initially identified. In the following studies, 42 replications and 196 unrelated studies were excluded, while 46 studies had full commentary. Subsequently, 38 articles were excluded as they were reviews (14), letters or conference abstracts (5), cell studies (15), and animal studies (4). Finally, 8 articles [18–25] were included. All studies were published between 2013 and 2018 and included a total of 723 CRC patients. The sample size ranged from 40 to 148 patients, who were all from Asia. Additionally, 4 of the studies [18–21] were published in English while 4 were [22–25] in Chinese. All studies were retrospective studies, with males accounting for 55–62.2%. GOLPH3 expression was determined by immunohistochemistry in all studies, and the thresholds for distinguishing GOLPH3 levels were found to be significantly different in each study. According to the Newcastle-Ottawa Scale, all studies scored above 5 (Table 1).

### GOLPH3 expression in CRC

Eight studies [18–25] provided data on GOLPH3 expression in 723 CRC tissue samples as well as in 401 adjacent normal colorectal tissue samples (Table 2). By conducting a random effects model ($I^2$ = 0.0%, $P$ = 0.453), the overall OR was observed to be 2.63 (95%CI = 1.96–3.52, $P<$0.001) (Fig 2), These results indicate that GOLPH3 is highly expressed in colorectal cancer tissues.

### GOLPH3 expression and clinicopathological parameter

All studies reported GOLPH3 expression in CRC and reported data for at least four clinicopathological parameters (Table 2). More specifically, 7 studies reported on gender; 7 studies reported on age; 7 studies reported on tumor stage; 6 studies reported on tumor differentiation; 8 studies reported on lymphatic metastasis; 6 studies reported on tumor size; and 4 studies reported on T stage (Table 2). When the data were combined separately, the high expression of GOLPH3 was significantly correlated with the clinicopathological parameters of the tumor. Specifically, the results were as follows: gender (male vs female) (OR = 0.89,95% CI = 0.69–1.16, $P$ = 0.381) (Fig 3A); age ($<$cut-off vs $\geq$cut-off) (OR = 0.89, 95%CI = 0.65–1.23, $P$ = 0.481) (Fig 3B); tumor stage (stage I/II vs stage III/IV) (OR = 0.73, 95%CI = 0.55–0.97,

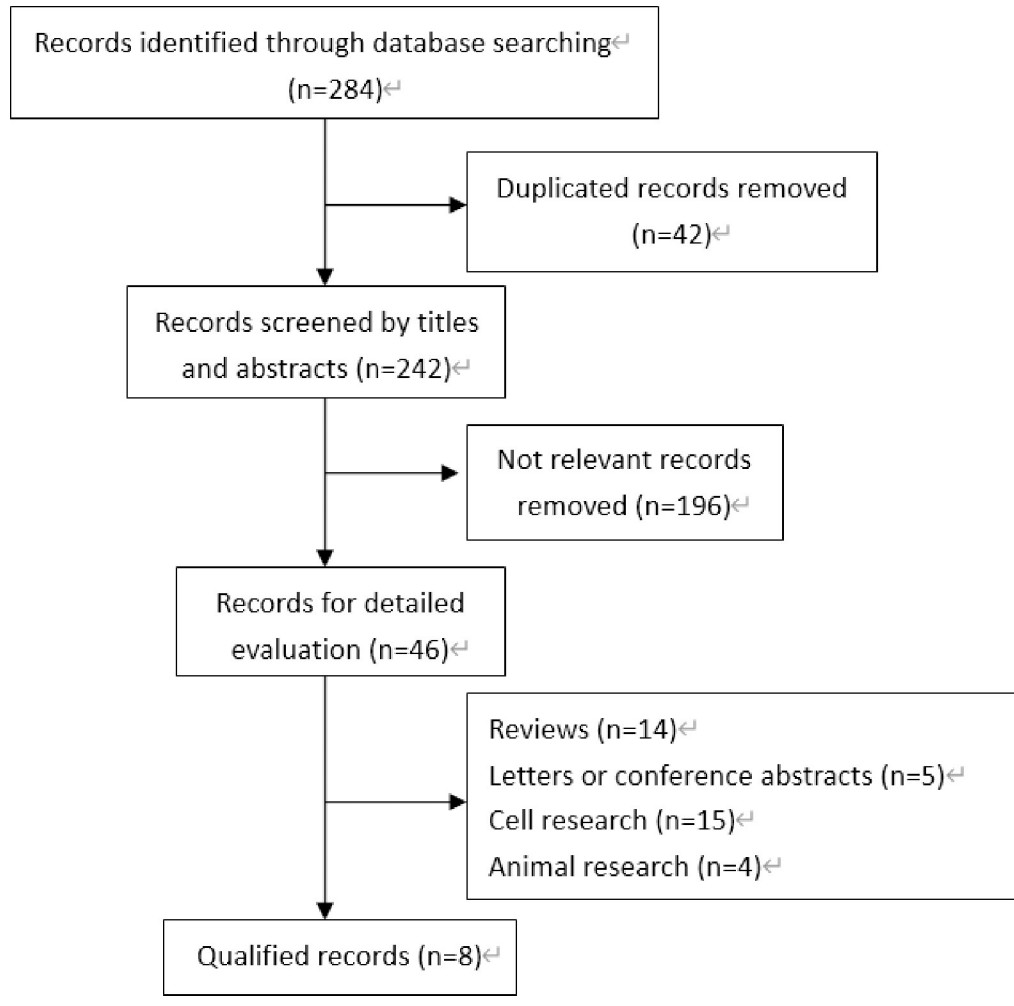

**Fig 1.** Flow diagram of study selection.

$P$ = 0.027) (Fig 3C); tumor differentiation (well differentiation vs moderate to poor differentiation) (OR = 0.56, 95%CI = 0.30–1.06, $P$ = 0.076) (Fig 3D); lymphatic metastasis (with lymphatic metastasis vs. without lymphatic metastasis) (OR = 1.27, 95%CI = 0.92–1.76, $P$ = 0.152) (Fig 3E); tumor size (<cut-off vs ≥ cut-off) (OR = 1.27, 95%CI = 0.86–1.88, $P$ = 0.229) (Fig 3F); and T stage (T1/2 vs T3/4) (OR = 0.70, 95%CI = 0.43–1.13, $P$ = 0.141) (Fig 3G).

**Table 1. Baseline characteristics of included studies.**

| Author | Year | Country | No. of cases | Male (%) | Clinical stage of patients | score |
|---|---|---|---|---|---|---|
| B Zhou | 2016 | China | 98 | 61(62.2%) | stageII-III | 7 |
| YT Guo | 2015 | China | 62 | - | stageI-III | 5 |
| KL Zhu | 2016 | China | 148 | 89(60.14%) | - | 6 |
| ZZ Wang | 2014 | China | 130 | 80(61.5) | stageII-IV | 6 |
| WL Xu | 2017 | China | 60 | 37(61.67%) | stageI-IV | 6 |
| SH Liu | 2018 | China | 40 | 22(55%) | stageI-IV | 6 |
| XF Yang | 2014 | China | 123 | 72(58.5) | stageI-IV | 6 |
| WS Yu | 2013 | China | 62 | 38(61.3%) | stageI-III | 6 |

**Table 2. The expression of GOLPH3 in cancer tissues and adjacent normal tissues and clinicopathological characteristics of patients with colorectal cancer.**

Expression of GOLPH3 (positive /all) (N)

| Author | Normal tissues | Cancer tissues | Gender | | Age (years) | | | Tumor stage | | Tumor differentiation | | Lymphatic metastasis | | Tumor size | | | | |
|---|---|---|---|---|---|---|---|---|---|---|---|---|---|---|---|---|---|---|
| | | | Male | Female | <cut-off | ≥cut-off | cut-off value | I-II | III-IV | Well | Moderate to poor | Yes | No | <cut-off | ≥cut off | cut-off value | T1-T2 | T3-T4 |
| B Zhou | 0/15 | 69/98 | 43/61 | 26/37 | - | - | - | 23/42 | 45/56 | 27/45 | 42/53 | 43/55 | 26/43 | - | - | - | 22/40 | 46/'58 |
| YT Guo | 15/62 | 33/62 | - | - | 17/30 | 16/32 | 60 | 13/33 | 20/29 | - | - | 20/29 | 13/33 | 23/40 | 10/22 | 5cm | - | - |
| KL Zhu | 11/87 | 77/148 | 48/89 | 29/59 | 32/67 | 45/81 | 65 | - | - | 35/82 | 42/66 | 55/94 | 22/54 | - | - | - | - | 77/148 |
| ZZ Wang | 10/75 | 56/130 | 27/80 | 29/50 | 26/58 | 30/72 | 60 | 24/45 | 32/85 | 7/17 | 44/104 | 30/80 | 26/50 | 19/43 | 37/87 | 4cm | 6/13 | 50/117 |
| WL Xu | 14/60 | 31/60 | 19/37 | 12/23 | 15/31 | 16/29 | 60 | 13/32 | 18/28 | 8/11 | 23/49 | 19/28 | 12/32 | 22/38 | 7/22 | 5cm | - | - |
| SH Liu | 14/40 | 26/40 | 14/22 | 12/18 | 7/15 | 19/25 | 60 | 12/25 | 14/15 | 8/19 | 18/21 | 20/21 | 6/19 | 18/28 | 8/12 | 5cm | - | - |
| XF Yang | - | 74/123 | 41/72 | 33/51 | 25/43 | 49/80 | 60 | 23/50 | 51/73 | 9/18 | 65/105 | 29/59 | 45/64 | 47/79 | 23/36 | 5cm | 6/19 | 68/104 |
| WS Yu | 15/62 | 33/62 | 19/38 | 14/24 | 17/30 | 16/32 | 60 | 13/33 | 20/29 | - | - | 20/29 | 13/33 | 23/40 | 10/22 | 5cm | - | - |

## Prognostic value of GOLPH3 in malignant tumors

Overall, 5 studies [18–21, 24] comprised of 622 patients who all underwent surgery for CRC, reported on GOLPH3 expression and OS in CRC. Five of the studies [18–21, 24] were univariate while two [20, 21] were multivariate (Table 3). The HR was 1.19 (95% CI = 0.47–1.92, $P>0.05$; $I^2$ = 60.3%, $P$ = 0.039) in the univariate analysis (Fig 4A) and 1.03 (95% CI = -0.52–2.58, $P>0.05$; $I^2$ = 50.1%, $P$ = 0.157) in the multivariate analysis (Fig 4B). Four studies [18–20,

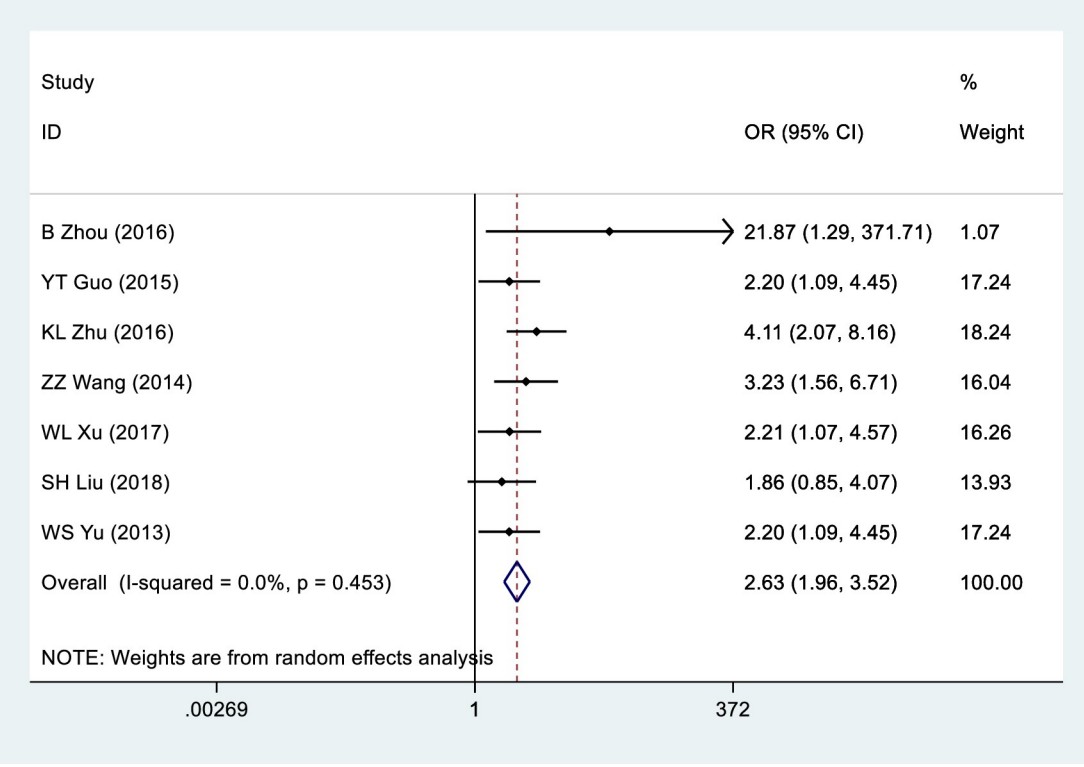

| Study ID | | OR (95% CI) | % Weight |
|---|---|---|---|
| B Zhou (2016) | | 21.87 (1.29, 371.71) | 1.07 |
| YT Guo (2015) | | 2.20 (1.09, 4.45) | 17.24 |
| KL Zhu (2016) | | 4.11 (2.07, 8.16) | 18.24 |
| ZZ Wang (2014) | | 3.23 (1.56, 6.71) | 16.04 |
| WL Xu (2017) | | 2.21 (1.07, 4.57) | 16.26 |
| SH Liu (2018) | | 1.86 (0.85, 4.07) | 13.93 |
| WS Yu (2013) | | 2.20 (1.09, 4.45) | 17.24 |
| Overall (I-squared = 0.0%, p = 0.453) | | 2.63 (1.96, 3.52) | 100.00 |

NOTE: Weights are from random effects analysis

.00269    1    372

**Fig 2. Forest plot of GOLPH3 expression in CRC tissue samples and adjacent normal colorectal tissue samples.** The horizontal line represents the range in which the truth value of the study exists. The dots on the horizontal line represent the effect size of a single study, and the size of the dots represents the weight. Diamonds represent the result of the merger.

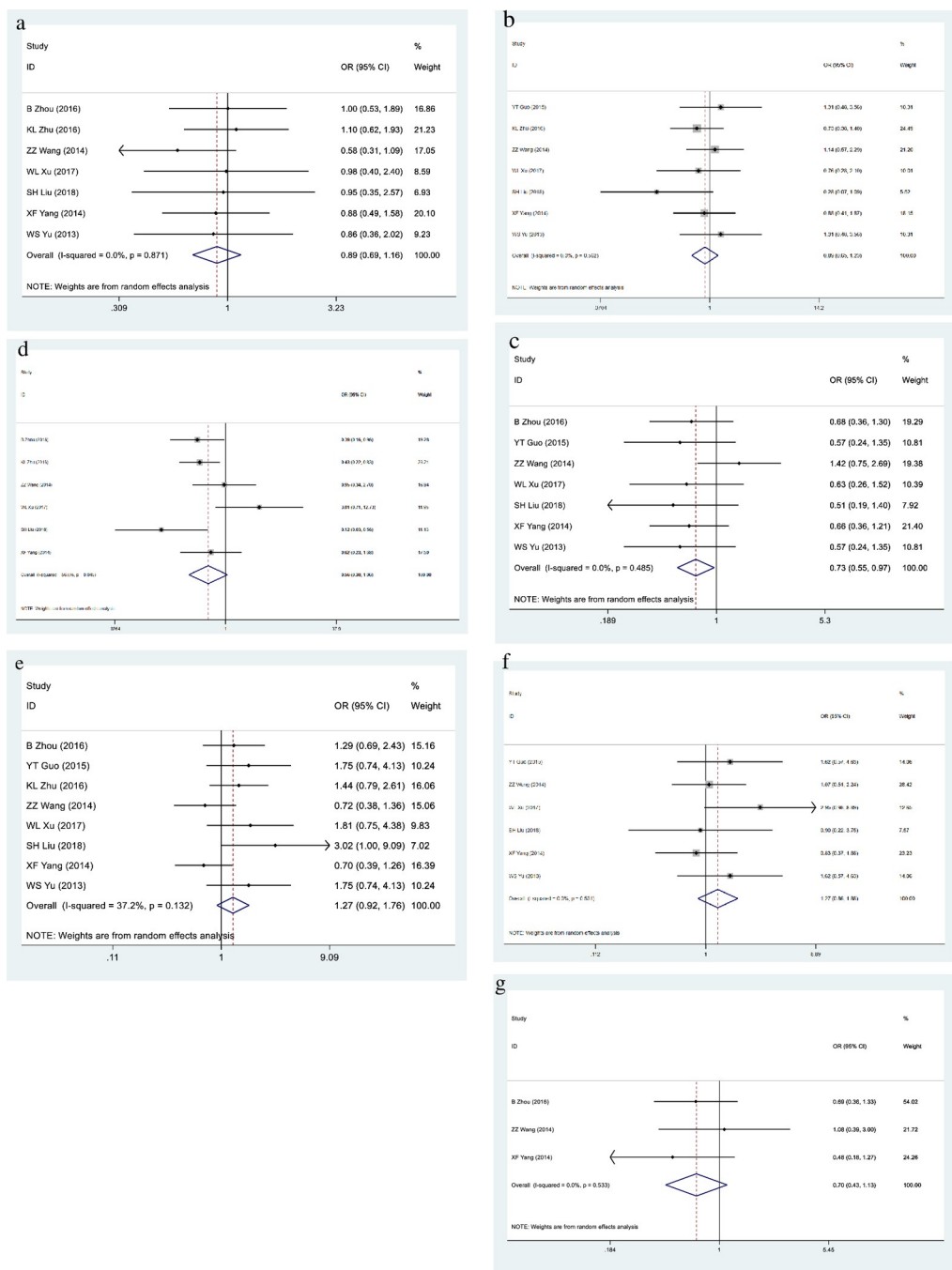

**Fig 3. The forest plot of ORs was assessed for association between GOLPH3 and clinicopathological parameter outcome.** a: gender (male vs female); b: age (<cut-off vs ≥cut-off); c: tumor stage (stage I/II vs stage III/IV); d: tumor differentiation (well differentiation vs moderate to poor differentiation); e: lymphatic metastasis (with lymphatic metastasis vs. without lymphatic metastasis); f: tumor size (<cut-off vs ≥ cut-off); g: T stage (T1/2 vs T3/4). The horizontal line represents the range in which the truth value of the study exists. The dots on the horizontal line represent the effect size of a single study, and the size of the dots represents the weight. Diamonds represent the result of the merger.

22] were obtained by KM curve. The results showed that high GOLPH3 expression was not significantly associated with poor OS prognosis.

**Table 3. Summary of HR with 95%CI for OS and DFS in each study.**

| Author | OS (HR with 95%CI) | | DFS (HR with 95%CI) | |
|---|---|---|---|---|
| | Univariate analysis | Multivariate analysis | Univariate analysis | Multivariate analysis |
| B Zhou (2016) | 1.898(0.837–4.301)/KM | - | - | - |
| YT Guo (2015) | 2.147(0.826–5.577)/KM | - | - | - |
| KL Zhu (2016) | 1.569(0.888–2.776)/KM | 2.354(1.237–6.152)/Rep | 1.433(0.799–2.568)/KM | 2.624(1.235–6.541)/Rep |
| ZZ Wang (2014) | 0.458(0.245–0.853)/Rep | 0.557(0.292–1.062)/Rep | 0.334(0.159–0.702)/Rep | 0.468(0.222–0.987)/Rep |
| XF Yang (2014) | 1.283(0.609–2.704)/KM | - | - | - |

Rep: Reported in the included studies; KM: Calculated from Kaplan-Meier curve.

In addition, 2 studies [20, 21] made up of 278 CRC patients reported HR with 95% CI for disease-free survival (DFS) in the univariate and multivariate models (Table 3). The pooled HR was 0.80 (95% CI = -0.26–1.86, p>0.05; $I^2$ = 81.5%, $P$ = 0.020) in the univariate analysis (Fig 5A) and 1.13 (95% CI = -0.82–3.08, p>0.05; $I^2$ = 59.8%, $P$ = 0.115) in the multivariate analysis (Fig 5B). The above results indicate that the high expression of GOLPH3 has no significant correlation with DFS.

## Discussion

CRC is a very common gastrointestinal malignancy. However, due to improvements in living standards as well as changes in diet, the incidence and mortality of CRC rise annually, which seriously influences quality of life. For the treatment of CRC, surgery is considered radical treatment, while adjuvant treatment is comprised of radiotherapy and chemotherapy. Once the recurrence and metastasis will increase the difficulty of treatment. Accordingly, to clarify the pathogenesis of colorectal cancer is of great significance for the early diagnosis, treatment and prognosis of colorectal cancer [26]. During malignant tumor proliferation, the activation of various proto-oncogenes or the inactivation of tumor suppressor genes is often accompanied, leading to the abnormal expression of proteins in tumor cells [27, 28]. Therefore, studying protein expression changes in tumor genesis is important in tumor targeted therapy and prognosis.

The GOLPH3 gene has been shown to promote tumorigenesis. GOLPH3 important for maintaining the maintenance of the Golgi ribbon structure, vesicle transport and Golgi glycosylation [29, 30]. The role of GOLPH3 in tumorigenesis may be related to the following cellular activities: regulation of Golgi-to-plasma membrane transport and acceleration of malignant secretory phenotypes [31, 32]; control of the internalization and circulation of key signaling molecules or increase of the glycosylation of cancer-associated glycoproteins [33, 34]; and the influence of DNA damage response and maintenance of genomic stability [35, 36]. GOLPH3 can promote the proliferation and inhibit the apoptosis of cancer cells via activation of the Wnt signaling pathway and can enhance the expression of β-catenin [37, 38]. Further studies have shown that GOLPH3 can activate AKT and AKT activated Wnt signaling through GSK-3β [39]. GOLPH3 is associated with the prognosis of CRC patients receiving 5-FU-based adjuvant chemotherapy following surgery and may serve as a potential predictor of 5-FU chemical sensitivity [21]. Centromere protein H interacts with GOLPH3 to inhibit the malignant phenotype of CRC in the mTOR signaling pathway and attenuate mTORC1 and mTORC2, thereby hindering the occurrence and development of CRC [40]. Numerous studies have confirmed that the expression level of GOLPH3 protein in various solid tumor tissues, such as esophageal cancer, gastric cancer, lung cancer, liver cancer and prostate cancer, is significantly higher than that in normal tissues.

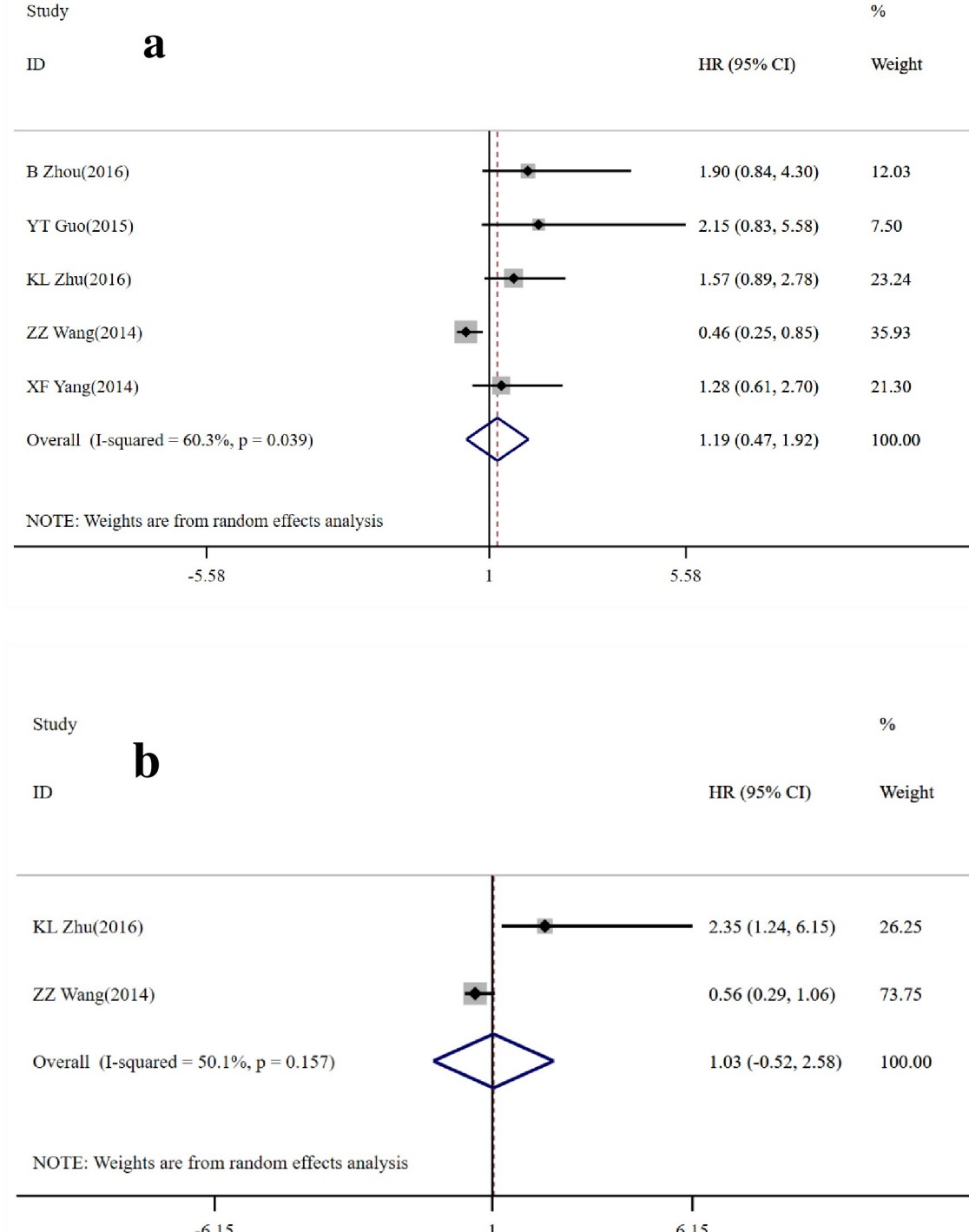

**Fig 4. Forest map of the relationship between GOLPH3 expression and overall survival (OS).** a: univariate analysis; b: multivariate analysis. The horizontal line represents the range in which the truth value of the study exists. The dots on the horizontal line represent the effect size of a single study, and the size of the dots represents the weight. Diamonds represent the result of the merger.

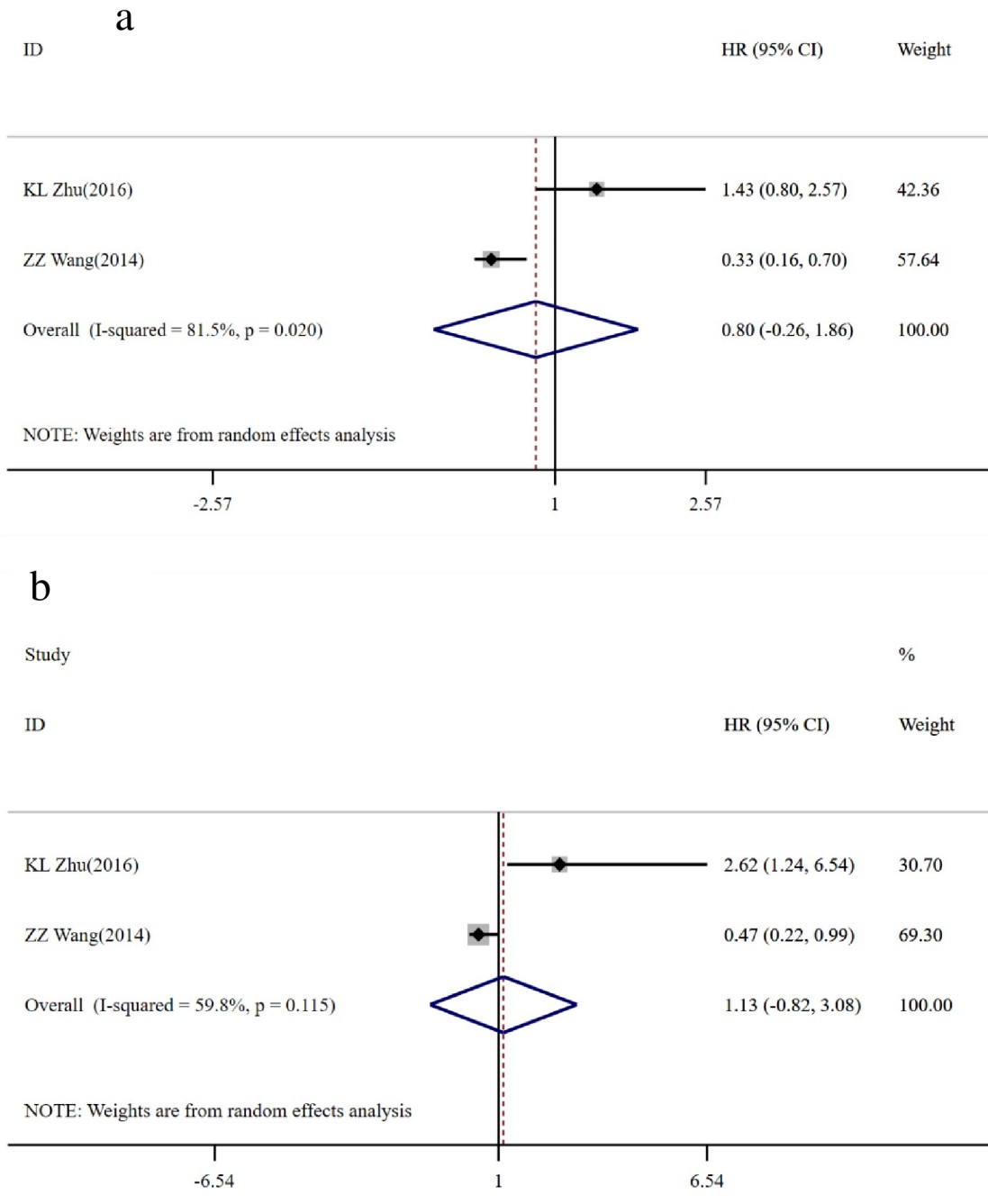

**Fig 5. Forest map of GOLPH3 expression and disease-free survival (DFS).** a: univariate analysis; b: multivariate analysis. The horizontal line represents the range in which the truth value of the study exists. The dots on the horizontal line represent the effect size of a single study, and the size of the dots represents the weight. Diamonds represent the result of the merger.

The results of this study demonstrated that the overall expression of GOLPH3 was found to be significantly higher than that of para-carcinoma tissue, which signifies that the expression of GOLPH3 in colorectal carcinoma was negatively associated with cell apoptosis. In addition, this study also found that the expression of GOLPH3 in CRC was related to clinical stage

rather than gender, age, differentiation degree, lymphatic metastasis, tumor size and T stage. The positive expression rate of GOLPH3 in patients with advanced stage was observed to be significantly higher than that in patients at an early stage [18, 19]. Moreover, studies [20, 41] have demonstrated that downregulation of GOLPH3 expression can inhibit the proliferation, invasion and migration of colorectal carcinoma, indicating that GOLPH3 plays an important role in the development of CRC. This study also analyzed the expression of GOLPH3 in CRC along with its clinical significance, providing guidance for the clinical treatment of CRC. As a result, methods to inhibit CRC may be determined by studying the related pathway of GOLPH3. This study also showed that the OS and DFS of CRC patients with positive expressions of GOLPH3 does not differ from that of patients with negative expressions. In light of the results of this study, it is suggested that the high expression of GOLPH3 is a factor of poor clinical stage in patients with CRC. As a result, GOLPH3 cannot be used as a prognostic indicator of CRC, and GOLPH3 may be not involved in the occurrence, development, invasion, metastasis and recurrence of CRC, which may be due to a small sample size.

Although we strived to conduct a comprehensive and scientific meta-analysis, this study has some limitations. First, all included studies were retrospective and were conducted in Asia with small sample sizes, which may lead to potential selection bias. Additionally, a question was put forward concerning the external validity of the results as well as their applicability to patients in Western countries. Therefore, prospective, large-sample, multinational studies should be performed in the future. Second, in regard to the eligible studies, the cut-off values of high expression/positive expression of GOLPH3 were different, and this inconsistency may serve as one of the reasons for the results described in this paper. Hence, further multicenter studies using standardized methods should be conducted. Third, the survival data of 4 studies were extracted from a Kaplan-Meier curve, which may affect the reliability of the prognosis. Finally, sensitivity analysis and publication bias cannot be accurately assessed due to the small number of literatures. However, random effects models were designed in order to reduce this outcome. Given the above limitations, the corresponding results should be interpreted with caution; likewise, the conclusions of this meta-analysis should be carefully drawn.

## Conclusions

This study demonstrated that the overexpression of GOLPH3 was found to be positively correlated with tumor stage, which is an adverse clinicopathological characteristic of CRC. Furthermore, it was not correlated with other adverse clinicopathological characteristics, such as lymphatic metastasis, tumor size, poor differentiation of tumor or T stage. As a result, GOLPH3 can not serve as a prognostic biomarker for CRC patients.

## Supporting information

**S1 Checklist. PRISMA 2009 checklist.**
(DOCX)

## Acknowledgments

The authors are pleased to acknowledge the suggestions from Tao Wang, which is important for manuscript revision.

## Author Contributions

**Conceptualization:** Tao Wang.

**Data curation:** Tao Wang, Jiandong Fei.

**Formal analysis:** Tao Wang, Jiandong Fei.

**Investigation:** Tao Wang.

**Project administration:** Shuangfa Nie.

**Software:** Tao Wang.

**Supervision:** Tao Wang.

**Visualization:** Shuangfa Nie.

**Writing – original draft:** Shuangfa Nie.

**Writing – review & editing:** Tao Wang.

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
