## [Decision Letter · Decision Letter 0]

20 Aug 2021

PONE-D-21-18847

Clinicopathologic and Prognostic Implications of Golgi Phosphoprotein 3 in Colorectal Cancer: A Meta-analysis

PLOS ONE

Dear Dr. Wang,

Thank you for submitting your manuscript to PLOS ONE. After careful consideration, we feel that it has merit but does not fully meet PLOS ONE’s publication criteria as it currently stands. Therefore, we invite you to submit a revised version of the manuscript that addresses the points raised during the review process.

We look forward to receiving your revised manuscript.

Kind regards,

Ajay Goel

Academic Editor

PLOS ONE

2. Thank you for submitting the above manuscript to PLOS ONE. During our internal evaluation of the manuscript, we found significant text overlap between your submission and the following previously published works, some of which you are an author.

https://www.jcancer.org/v10p5754.pdf

Please revise the manuscript to rephrase the duplicated text, cite your sources, and provide details as to how the current manuscript advances on previous work. Please note that further consideration is dependent on the submission of a manuscript that addresses these concerns about the overlap in text with published work.

We will carefully review your manuscript upon resubmission, so please ensure that your revision is thorough

Reviewers' comments:

Reviewer's Responses to Questions

**Comments to the Author**

1. Is the manuscript technically sound, and do the data support the conclusions?

Reviewer #1: Yes

2. Has the statistical analysis been performed appropriately and rigorously? 

Reviewer #1: Yes

3. Have the authors made all data underlying the findings in their manuscript fully available?

Reviewer #1: Yes

4. Is the manuscript presented in an intelligible fashion and written in standard English?

Reviewer #1: Yes

5. Review Comments to the Author

Reviewer #1: The authors performed a meta-analysis to assess its association with the clinicopathological characteristics of patients and evaluate the prognostic significance of GOLPH3 in CRC. GOLPH3 was found to be highly expressed in tumor tissues compared to that of adjacent colorectal tissues (OR, 2.63), and overexpression of GOLPH3 was observed to be significantly correlated with advanced clinical stage (OR, 3.42). The pooled analysis showed that GOLPH3 overexpression was not associated with worse overall survival (OS) (HR=1.14, 95% CI: 0.42-1.86, p>0.05) and disease-free survival (DFS) (HR=0.80, 95% CI:-0.26-1.86, p>0.05).

Comments:

1. Only Chinese cohort

All datasets are derived from only China. It may be difficult to include additional dataset, it might be better to analyze datasets from various countries.

2. Figure legends

I could not find Figure legends. Please put Figure legends.

3. Title of each figure

Figure 3-13 looks similar, and reader of journal will not be able to recognize which parameter you analyzed. Please put the title which show the target of analysis.

4. Figure 10

In Figure 10, I2, p-value, HR are different from that in Text. Please verify them.

Overexpression of GOLPH3 was observed to be significantly correlated with only advanced clinical stage. So, I think that GOLPH3 can not be a biomarker in CRC. Even though the data itself is not so interesting in this manuscript, many researchers may be released from the analysis of GOLPH3.

6. PLOS authors have the option to publish the peer review history of their article (what does this mean?). If published, this will include your full peer review and any attached files.

Reviewer #1: No

---

## [Author Response · Author response to Decision Letter 0]

23 Sep 2021

Dear Reviewers:

Thank you very much for your review comments on my paper “P Clinicopathologic and prognostic implications of Golgi Phosphoprotein 3 in colorectal cancer: a meta-analysis”. Each of your review comments has brought great help to my scientific research work. The manuscript has been revised in accordance with the reviewer’s comments and there are traces of changes in the marked-up copy. This is a rebuttal letter that responds to each point raised by the academic editor and reviewer(s).

Additional requirements:

https://journals.plos.org/plosone/s/file?id=wjVg/PLOSOne_formatting_sample_main_body.pdf andhttps://journals.plos.org/plosone/s/file?id=ba62/PLOSOne_formatting_sample_title_authors_affiliations.pdf

Response: I logged into the website and downloaded the sample. Then, modify the manuscript according to the sample.

2. Thank you for submitting the above manuscript to PLOS ONE. During our internal evaluation of the manuscript, we found significant text overlap between your submission and the following previously published works, some of which you are an author.https://www.jcancer.org/v10p5754.pdfWe would like to make you aware that copying extracts from previous publications, especially outside the methods section, word-for-word is unacceptable. In addition, the reproduction of text from published reports has implications for the copyright that may apply to the publications. Please revise the manuscript to rephrase the duplicated text, cite your sources, and provide details as to how the current manuscript advances on orevious work. Please note that further consideration is dependent on the submission of a manuscript that addresses these concerns about the overlap in text with published work. We will carefully review your manuscript upon resubmission, so please ensure that your revision is thorough.

Response: I have revised the manuscript thoroughly to rephrase the duplicated text, cite my sources, and provide details as to how the current manuscript advances on orevious work in"Authors’ contributions". Finally, I checked the revised article through "TurnitinUK" and found that the repetition rate was less than 30%.

3. Please review your reference list to ensure that it is complete and correct. If you have cited papers that have been retracted, please include the rationale for doing so in the manuscript text, or remove these references and replace them with relevant current eferences. Any changes to the reference list should be mentioned in the rebuttal letter that accompanies your revised manuscript. If you need to cite a retracted article, indicate the article's retracted status in the References list and also include a citation and ful reference for the retraction notice. 

Response: I have checked the references in the article and am quite sure that they are complete and correct. I didn't cite the retracted paper.

Response: It has been modified as required.

Review Comments to the Author：

Reviewer #1:The authors performed a meta-analysis to assess its association with the clinicopathological characteristics of patients and evaluate the prognostic significance of GOLPH3 in CRC. GOLPH3 was found to be highly expressed in tumor tissues compared to that of adjacent colorectal tissues (OR, 2.63), and overexpression of GOLPH3 was observed to be significantly correlated with advanced clinical stage OR, 3.42). The pooled analysis showed that GOLPH3 overexpression was not associated with worse overall survival (OS) (HR=1.14, 95% Cl: 0.42-1.86, p>0.05) and disease-free survival (DFS) (HR-0.80, 95% Ci:-0.261.86, p>0.05). 

Comments:1. Only Chinese cohortAll datasets are derived from only China. It may be, difficult to include additional dataset, it might be better to analyze datasets from various countries.

Response: Two other authors and I searched English databases, such as PubMed, Cochrane Library, Web of Science, Medline, Embase, and Chinese databases, such as CNKI and WanFang Databases, respectively, to find literature that meets the criteria. Our search was comprehensive and scientific, but unfortunately we did not find any English literature that met the standard after screening. Our data are real and reliable, and the outcome is objective and scientific. Of course, we are looking forward to prospective, large-sample, multinational studies in the future.

Comments:2. Figure legendsI could not find Figure legends. Please put Figure legends

Response: It has been modified as required.

Comments:3. Title of each figureFigure 3-13 looks similar, and reader of journal will not be able to recognize which parameter you analyzed. Please put the title which show the target of analysis. 

Response: It has been modified as required, as follows:

Fig 3. The forest plot of ORs was assessed for association between GOLPH3 and clinicopathological parameter outcome. a : gender(male vs female);b : age(<cut-off vs ≥cut-off);c : tumor stage (stage I/II vs stage III/IV);d : tumor differentiation (well differentiation vs moderate to poor differentiation);e : lymphatic metastasis (with lymphatic metastasis vs. without lymphatic metastasis);f : tumor size (<cut-off vs ≥ cut-off);g : T stage (T1/2 vs T3/4). The horizontal line represents the range in which the truth value of the study exists. The dots on the horizontal line represent the effect size of a single study, and the size of the dots represents the weight. Diamonds represent the result of the merger.

Fig 4. Forest map of the relationship between GOLPH3 expression and overall survival (OS). a : univariate analysis; b : multivariate analysis. The horizontal line represents the range in which the truth value of the study exists. The dots on the horizontal line represent the effect size of a single study, and the size of the dots represents the weight. Diamonds represent the result of the merger

Fig 5. Forest map of GOLPH3 expression and disease-free survival (DFS). a : univariate analysis; b : multivariate analysis. The horizontal line represents the range in which the truth value of the study exists. The dots on the horizontal line represent the effect size of a single study, and the size of the dots represents the weight. Diamonds represent the result of the merger. 

Comments:4. Figure 10. In Figure 10, 12, p-value, HR are different from that in Text. Please verify them. 

Response: I have carefully checked all the results in the article and corrected the mistakes. Thank you for your carefulness and reminder.

---

## [Decision Letter · Decision Letter 1]

2 Nov 2021

Clinicopathologic and Prognostic Implications of Golgi Phosphoprotein 3 in Colorectal Cancer: A Meta-analysis

PONE-D-21-18847R1

Dear Dr. Wang,

We’re pleased to inform you that your manuscript has been judged scientifically suitable for publication and will be formally accepted for publication once it meets all outstanding technical requirements.

Kind regards,

Ajay Goel

Academic Editor

PLOS ONE

Additional Editor Comments (optional):

Reviewers' comments:

Reviewer's Responses to Questions

**Comments to the Author**

1. If the authors have adequately addressed your comments raised in a previous round of review and you feel that this manuscript is now acceptable for publication, you may indicate that here to bypass the “Comments to the Author” section, enter your conflict of interest statement in the “Confidential to Editor” section, and submit your "Accept" recommendation.

Reviewer #1: All comments have been addressed

2. Is the manuscript technically sound, and do the data support the conclusions?

Reviewer #1: Yes

3. Has the statistical analysis been performed appropriately and rigorously? 

Reviewer #1: Yes

4. Have the authors made all data underlying the findings in their manuscript fully available?

Reviewer #1: Yes

5. Is the manuscript presented in an intelligible fashion and written in standard English?

Reviewer #1: Yes

6. Review Comments to the Author

Reviewer #1: I'm satisfied with response from the author. I have no additional comments. The revised manuscript looks better than the original version.

7. PLOS authors have the option to publish the peer review history of their article (what does this mean?). If published, this will include your full peer review and any attached files.

Reviewer #1: No

---

## [Editor Report · Acceptance letter]

11 Nov 2021

PONE-D-21-18847R1 

Clinicopathologic and Prognostic Implications of Golgi Phosphoprotein 3 in Colorectal Cancer: A Meta-analysis 

Dear Dr. Wang:

I'm pleased to inform you that your manuscript has been deemed suitable for publication in PLOS ONE. Congratulations! Your manuscript is now with our production department. 

Kind regards, 

on behalf of

Dr. Ajay Goel 

Academic Editor

PLOS ONE